# The relationship between heart rate variability and glucose clearance in healthy men and women

Abigail Nickel[ORCID]*, Robert Buresh[ORCID], Cherilyn McLester, Andre Canino, Gabe Wilner, Keilah Vaughan, Pedro Chung, Brian Kliszczewicz[ORCID]*

Department of Exercise Science and Sport Management, Kennesaw State University, Kennesaw, Georgia, United States of America

* amarmuro@students.kennesaw.edu (AN); bkliszcz@kennesaw.edu (BK)

## Abstract

Heart rate variability (HRV) is a non-invasive indicator of the activity of the autonomic nervous system, which regulates many physiological functions including metabolism. The purpose of this study was to quantify the relationship between resting markers of HRV and oral glucose tolerance test (OGTT) response. Eighteen healthy individuals (10 males, 8 females, (23.8±2.9 years) underwent a 10-minute resting HRV recording. The final five minutes were evaluated via Kubios HRV Standard for: root mean square of successive differences (RMSSD), standard deviation of normal-to-normal sinus beats (SDNN), high frequency (HF), and low frequency (LF). A standard 2-hour OGTT was then administered. Glucose was measured via finger stick before, 30-minutes post, 1-hour post, and 2-hours post OGTT. Pearson correlations demonstrated that RMSSD, SDNN, HF and LF were strongly correlated to fasting blood glucose (FBG) for the group (p<0.05) but not for glucose area under the curve (AUC). When analyzed by sex, only males demonstrated significant correlations between AUC and RMSSD, SDNN, and LF (p<0.05). An independent samples t-test revealed no sex differences for FBG, AUC, RMSSD, SDNN, HF and LF. These findings provide new and interesting insights into the relationship of autonomic activity and glucose uptake, highlighting sex-based relationships.

## Introduction

The continuous rise in the prevalence of type 2 diabetes globally [1] warrants the development of new and innovative approaches to identify evidence of metabolic dysfunction, and to explore connections with other systems that may lead to chronic conditions (e.g., cardiovascular disease). As a general function of homeostasis, multiple systems respond to internal and external stimuli in order to maintain systemic norms (e.g., temperature, pH, glucose). Regulation of plasma glucose is one such system that is susceptible to frequent fluctuations due to a multitude of autonomic nervous system (ANS) mediated factors such as intestinal absorption, glycogenolysis, and gluconeogenesis [2]. A more thorough understanding of these fluctuations

**Data Availability Statement:** Data are available from the BioStudies repository at DOI: 10.6019/S-BSST1348 (accession number S-BSST1348).

**Funding:** The authors received no specific funding for this work.

**Competing interests:** The authors have declared that no competing interests exist.

may prove beneficial in both applied and clinical settings, particularly if these fluctuations can be predictive.

The ANS has widespread innervation to nearly every organ system, with prominent effects on the cardiovascular and digestive systems through the balance of activity from the parasympathetic nervous system (PSNS) and sympathetic nervous system (SNS) divisions [3]. The ANS plays a key role in regulating a litany of physiological processes [4] and is considered a good indicator of physiological readiness and overall homeostatic gain (i.e., sensitivity to change) [5]. The ANS can be noninvasively evaluated through variations in cardiovascular activity referred to as heart rate variability (HRV) [6]. This is accomplished through the measurement of timing between beat-to-beat intervals using an electrocardiogram or a beat-to-beat detecting device. Recently, HRV has been used as a tool for a variety of applications ranging from a prognostic indicator for cardiovascular conditions [7], to predictions of peek aerobic performance and even recovery status for exercise training [8].

One of the most common tests used to evaluate metabolic function and glucose control is the 2-hour oral glucose tolerance test (OGTT). This test involves the consumption of a standardized amount of glucose and measurements of blood glucose concentration to allow for the assessment of glucose clearance at various time points over a two-hour period. In a recent study performed by Prasertsri et. al, HRV was shown to be acutely altered with the consumption of a high glucose beverage, demonstrating a relationship between the two measures; however, the authors did not speculate as to the cause or extent of this relationship [9]. In healthy individuals, the ANS can exert a number of influences on glucose regulation. For instance, PSNS activity promotes insulin secretion and subsequent reduction of blood glucose [10], while SNS activity mobilizes glucose resulting in the elevation of blood glucose [11]. Importantly, HRV is a noninvasive measure of ANS activity, and may prove to be an effective tool in assessing and assisting in the explanation of variations in glucose regulation. Therefore, the purpose of this study was to evaluate the predictive value of HRV on the results of an OGTT in healthy participants.

## Materials and methods

### Participants

Prior to data collection, the Institutional Review Board approved all testing procedures and protocols. Ten apparently healthy males (24.0 ± 2.0 years) and eight apparently healthy females (23.5 ± 3.6 years) participated in the current study. Each volunteer was made aware of the procedures and risks associated with the study and signed a written informed consent and completed a health history questionnaire. Exclusion criteria included those who were pregnant, missing a limb, reported having cardiovascular, pulmonary, or metabolic conditions, had a body mass index (BMI) outside of the normal range (18.5–24.9 kg·m$^{-2}$) or were taking medication that interfered with metabolism. Inclusion and exclusion criteria were determined through self-report through the health history questionnaire. Recruitment took place via word of mouth primarily from the university community and the surrounding metropolitan area from June 30, 2022 until May 1, 2023. Prior to visiting the lab, participants were instructed to wear clothing that was light and comfortable, fast for a minimum of 12 hours (except water) and avoid exercise and alcohol for 24 hours leading up to the lab visit.

### Experimental design

Participants visited the institution's exercise physiology laboratory on one occasion and completed data collection between the hours of 7:00 am and 12:00 pm. Upon arrival to the lab, participants' height and weight was measured via stadiometer (WB-3000, Tanita, Tokyo, Japan).

Next, a body composition assessment via bioelectrical impedance analysis (BIA) (InBody 770, Cerritos, CA) was conducted. Participants were then taken back to a quiet, dimly lit room and instructed to lay in a supine position on an examination bed for 10-minutes while resting HRV was recorded on the Finapres NOVA (Finapres Medical Systems, Enschede, the Netherlands). Three ECG electrodes were placed inferior to the right clavicle, inferior to the left clavicle, and about two centimeters medial to the anterior superior iliac spine after applying abrasive and wiping the skin sites. After HRV collection, a urine sample was collected to confirm hydration status through urine specific gravity (USG). Immediately following was the measurement of baseline capillary blood glucose via finger stick and glucometer analysis (Bayer, Contour NextOne, Leverkusen, Germany). Participants were then instructed to consume entirely a standard OGTT beverage containing 75 grams of glucose (Trutol$^{TM}$) within a one-minute period. After consumption, the time was recorded to indicate the 30, 60, and 120-minute marks for the subsequent glucose readings. During each waiting period, participants were instructed to limit movement in order to minimize muscular activity and its known effects on blood glucose clearance. Once the final glucose reading was recorded, the visit was concluded, and participants were allowed to leave the laboratory.

## Heart rate variability analysis

Measurements for the current study were collected using the Finapres NOVA, and ECG files were transferred to Kubios software (version 3.0.5). The first five minutes of the 10-minute recording were discarded for acclimatation. The last five minutes of resting time points were analyzed for HRV and RHR. Kubios software was used to analyze time domain and frequency domain measures. Detrending was set to smoothing priors with a smoothing parameter of 500, lambda of 0.035 Hz, and Interpolation rate of 4 Hz.

To detect the presence of artifact or noise, the Kubios "low artifact correction" filter with a ± 0.35 sec sensitivity to R-R abnormalities compared to the local average was used [12–14].

The time and frequency domain measures of HRV chosen for the current study include the root mean square of successive differences (RMSSD), standard deviation of normal-to-normal sinus beats (SDNN), low frequency (LF) and high frequency (HF). LF and HF were determined through the Fast Fourier Transformation algorithm (FFT), which analyzed the spectrum of frequencies and separated them into the categories of LF (0.04–0.15 Hz) and HF (0.15–0.4 Hz). LF indicated both PNS and SNS activity, while HF and RMSSD represented PNS activity [15].

## Urine specific gravity

Participants provided a urine sample which was used to quantify hydration status via USG. Distilled water was dropped into the well on the refractometer to establish a zero point. Thereafter, a few drops of urine were applied into the well and analyzed, and USG was then recorded.

## Oral glucose tolerance test and area under the curve

Estimation of total area under the curve (AUC) for glucose tolerance was calculated from the OGTT results using Tai's Mathematical Model [16]. This model divides the total AUC into small individual segments whose areas can be precisely determined according to existing geometric formulas and then added together to obtain the total AUC as seen in Fig 1 [16]. Tai's formula was chosen for the current study as it allows glucose samples to be taken with differing time intervals while still precisely determining the total AUC and expressed as arbitrary units (AU) [16].

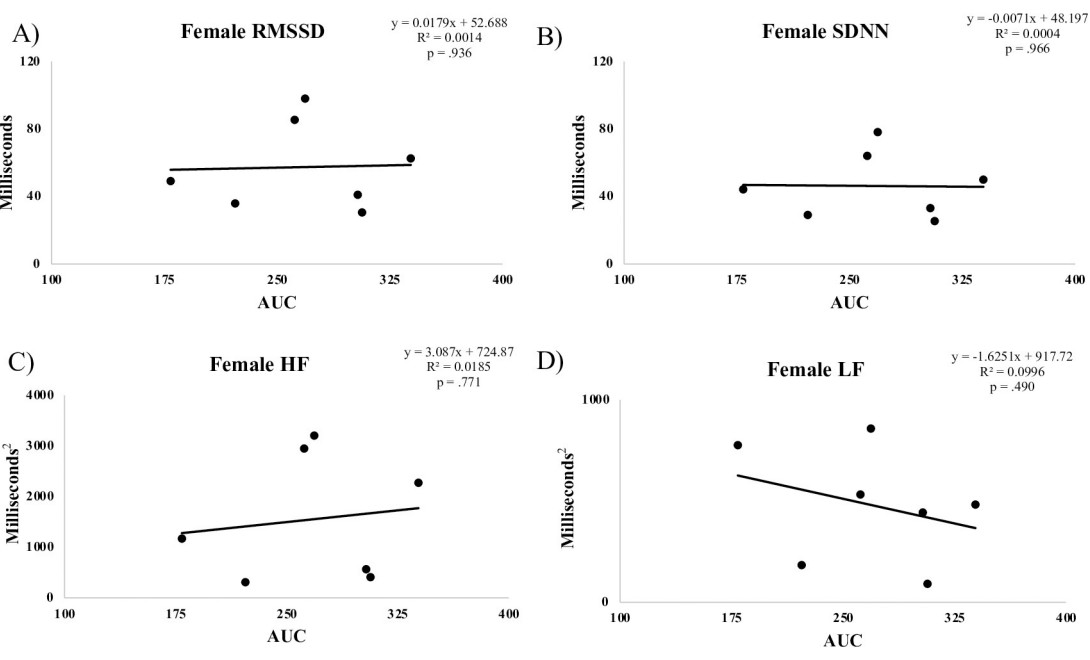

**Fig 1. Correlation between group AUC and HRV measures.**

## Menstrual cycle survey

Upon entering the lab, female participants were asked to record the estimated start date of their last known menstrual cycle. The estimated cycle phase was determined by counting back from the lab visit to the reported start of last known cycle (Table 1). Cycle phases were defined as menstruation (day 1–5), follicular (day 6–13), ovulation (day 14), and luteal (day 15–28) using a 28-day cycle [17].

## Statistical analysis

All data were entered and analyzed in SPSS version 28 software (Chicago, IL). Missing or excluded data were omitted listwise, i.e., observations with missing values on any of the variables in the analysis were omitted from analysis in SPSS. A Shapiro-Wilk Normality Test was performed on all HRV data and found that normality was not violated, and data did not require log transformation. Measurements of RMSSD (ms), SDNN (ms), LF (ms$^2$), HF (ms$^2$) were analyzed. To determine whether there was a relationship between HRV and OGTT, two-tailed Pearson product correlations were conducted between the AUC for glucose during OGTT and HRV measurements including HF, LF, SDNN and RMSSD. Data was then spilt by sex and two-tail Pearson product correlations were conducted between the AUC and HRV

**Table 1. Female participants in each estimated phase of their menstrual cycle.**

| Estimated menstrual cycle phase reported on V1 | Number of Participants | Days into Cycle |
|---|---|---|
| Menstruation (Day 1–5) | 1 | 5 |
| Follicular (Day 6–13) | 2 | 8 ± 1.4 |
| Ovulation (Day 14) | 0 | N/A |
| Luteal (Day 15–28) | 5 | 27.6 ± 5.4 |
| Total Mean/SD | 8 | 19.9 ± 11.5 |

**Table 2. Descriptive statistics.**

| Measure | All | Males | Females |
|---|---|---|---|
|  | (n = 15) | (n = 8) | (n = 7) |
| Height (cm) | 171.9 ± 10 | 178.7 ± 8.4 | 164.2 ± 6.8 |
| Weight (kg) | 66.5 ± 34 | 88.7 ± 8.8 | 69.8 ± 16.3 |
| Body Composition (% fat) | 22.9 ± 10 | 17.2 ± 6.5 | 29.4 ± 16.2 |
| Age (yrs) | 24.2 ± 2.8 | 24.3 ± 2.1 | 24.1 ±3.7 |

measurements including HF, LF, SDNN and RMSSD. A two-tailed Independent Samples T-test was used to determine sex-based differences between AUC, RMSSD, SDNN, HF, and LF. Alpha level was set at 0.05.

## Results

Eighteen apparently healthy volunteers completed the study: ten males and eight females. Two males and one female were removed from data analysis due to artifact or ECG recording error, 15 total participants were analyzed, eight males and seven females. Descriptive data can be seen in Table 2. Participant fasting times were; Total—13.1 +/- 1.8 hours, Males—12.6 +/- 2.1 hours, and Females—13.7 +/- 1.2 hours. All females reported a start of their last menstrual cycle within a time period suggestive of a normal menstrual cycle (Table 1). Group OGTT results can be seen in Fig 2. Results of the Pearson correlations demonstrated that RMSSD, SDNN, HF and LF were strongly correlated to fasting blood glucose (FBG) for the total group (p<0.05). The Pearson correlations can be seen in Table 3. There were no observed

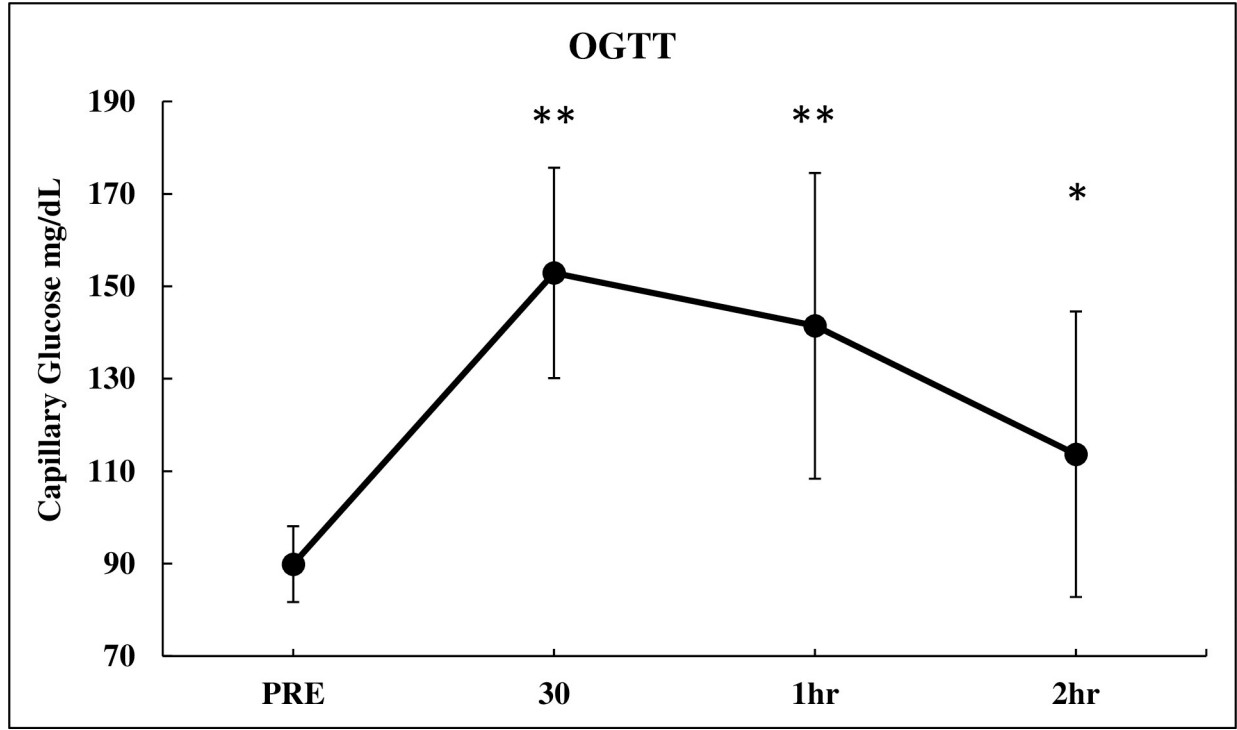

**Fig 2. Capillary blood glucose over time.** Values presented as means ± SD. Significantly different form PRE * = p < 0.05, ** = p < 0.001.

**Table 3. Pearson correlation coefficients.**

|  |  | AUC | FBG |
|---|---|---|---|
| **RMSSD (ms)** | Group | r = .365, p = .181 | **r = .512, p = .043**[*] |
|  | Male | **r = .736, p = .037**[*] | **r = .829, p = .006**[*] |
|  | Female | r = .038, p = .936 | r = -.022, p = .962 |
| **SDNN (ms)** | Group | r = .303, p = .273 | **r = .502, p = .048**[*] |
|  | Male | **r = .754, p = .031**[*] | **r = .903, p < .001**[*] |
|  | Female | r = -.020, p = .966 | r = -.068, p = .885 |
| HF (ms$^2$) | Group | r = .397, p = .142 | **r = .594, p = .015**[*] |
|  | Male | r = .690, p = .058 | **r = .916, p < .001**[*] |
|  | Female | r = .136, p = .771 | r = .009, p = .985 |
| LF (ms$^2$) | Group | r = .245, p = .378 | **r = .550, p = .027**[*] |
|  | Male | **r = .865, p = .006**[*] | **r = .703, p = .035**[*] |
|  | Female | r = -.316, p = .490 | r = -.115, p = .807 |
| **RHR (bpm)** | Group | r = .024, p = .933 | r = -.225, p = .421 |
|  | Male | r = -.357, p = .386 | r = -.322, p = .437 |
|  | Female | r = .470, p = .287 | r = .682, p = .092 |

AUC = Area under the curve, FBG = Fasting blood glucose, RMSSD = Root mean square of successive differences, SDNN = Standard deviation of normal sinus beats, HF = High frequency, LF = Low frequency.

[*] = Significant Correlation

correlations for AUC and RMSSD, SDNN, HF, LF and RHR for the group (p>0.05) (Table 3, Fig 1). When sexes were analyzed separately, we found significant correlations in males between AUC and RMSSD, SDNN, and LF (p<0.05), with a trend toward significance in males between AUC and HF (r = .690, p = .058) (Fig 3). No significant correlations between AUC and HF, RMSSD, RHR were observed in females (p>0.05) (Fig 4). Independent Samples T-test revealed no sex differences for AUC, RMSSD, SDNN, HF and LF (p>0.05) (Table 4).

## Discussion

The purpose of this study was to evaluate the predictive value of HRV on the results of an OGTT in healthy participants. The primary findings revealed that resting HRV demonstrated a strong positive correlation to the FBG measures for the total group. When accounting for sex, this relationship remained in males but was lost in females. When evaluating the relationship between HRV and glucose AUC derived from the OGTT, no relationships were observed among any of the assessed markers. When accounting for sex, only males showed a significant relationship between HRV and AUC.

The participants in this study were considered healthy, normal weight individuals with an average capillary FBG of 89.9 ± 8.2 mg/dL. Normal fasting glucose measures are well established to be between 80–100 mg/dL [18], which is a vital range to maintain proper pressure gradient to support glucose entry into tissue cells. The role of the ANS in glucose regulation is multifaceted, with variables that go beyond the scope of the current study but are worth mentioning as they may relate to the current findings. During periods of hypoglycemia, the mobilization of glycogen occurs in order to maintain appropriate glucose concentrations, with 70–90% of this response mediated by the ANS [19]. This response is believed to be mediated by both branches; however, early-stage hypoglycemia (75–85 mg/dL) results in PSNS activation of pancreatic α-cell islets [19] and subsequent glucagon secretion [20], although α-cell islets

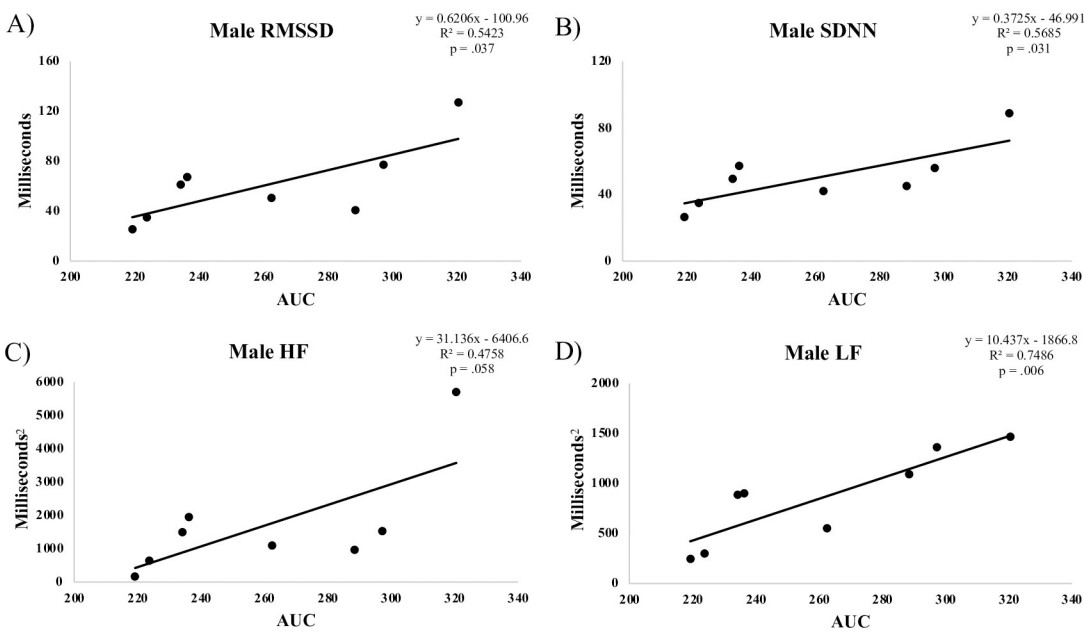

**Fig 3. Correlation between male AUC and HRV measures.**

recruitment increases once FBG is below 100 mg/dL [19]. Our study supports these notions by demonstrating a positive relationship between FBG and HRV markers RMSSD and HF, which are widely accepted markers of PSNS, presenting a possible rationale for the positive correlation observed.

It is well established that the SNS branch, when activated, results in the mobilization of glycogen via glucose elevating hormones–epinephrine, norepinephrine, glucagon, and growth

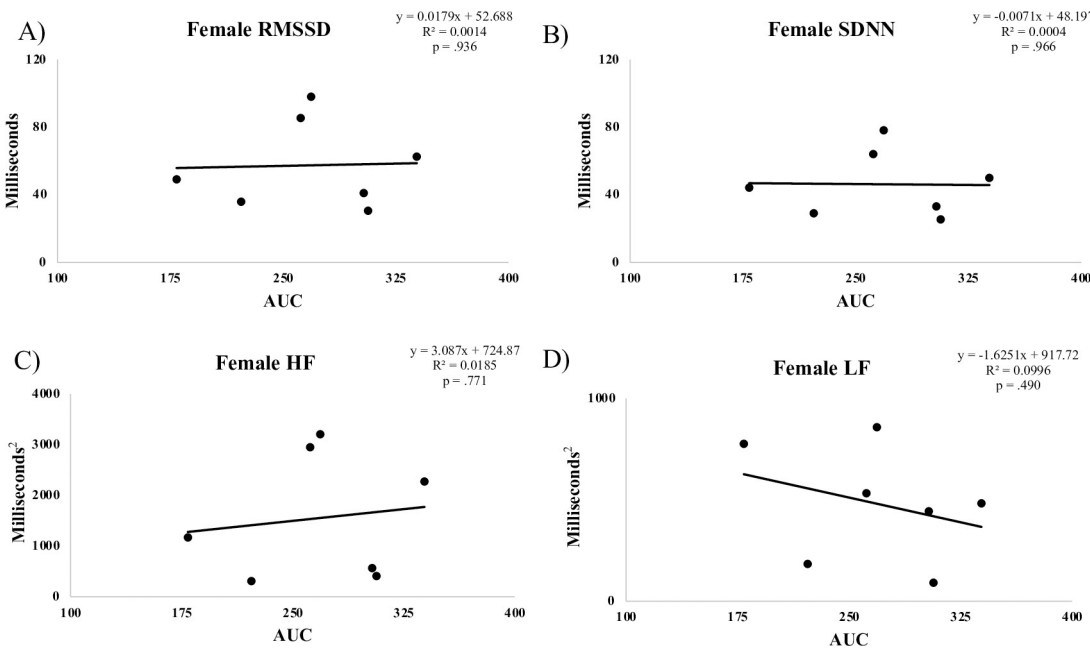

**Fig 4. Correlation between female AUC and HRV measures.**

Table 4. Independent samples t-test between males and females.

|  | Mean ± SD (M) | Mean ± SD (F) | p |
|---|---|---|---|
| RMSSD (ms) | 60.2 ± 34.6 | 57.5 ± 25.7 | .902 |
| SDNN (ms) | 50.3 ± 20.1 | 46.3 ± 19.4 | .748 |
| HF (ms$^2$) | 1736 ± 1843 | 1554 ± 1238 | .910 |
| LF (ms$^2$) | 877 ± 478 | 480 ± 280 | .075 |
| AUC (AU) | 258 ± 43 | 280 ± 52 | .438 |
| FBG (mg/dL) | 93.4 ± 9 | 86.1 ± 7 | .081 |

hormone [21]. It is generally believed that the elevation of SNS activity results in the concomitant measurable withdrawal of PSNS activity; however, previous findings in our lab found increases in circulating neurohormones epinephrine and norepinephrine, without changes in RMSSD or HF [22]. The findings of the current study are reflective of this, with SDNN and LF believed to be reflective of both PSNS and SNS activity, demonstrating a positive relationship with FBG. These results conflict with those of Rothberg et al. who demonstrated no relationships in resting HRV and FBG in healthy adults [23]. Furthermore, when this group looked at the same relationship in adults diagnosed with type 2 diabetes, a negative relationship was reported which is in contrast to the positive relationship in the current study. One potential reason for the discrepancy between the findings of our study and that of Rothberg et al. may be the body position in which HRV was recorded. Supine positions, which were used in this study, sometimes yield higher indices of vagal tone than seated postures [24], which may explain the differences. The negative relationship observed in the type 2 diabetic group may be related to the disease or the age of that population, and this, too, may contribute to the contrast in our findings.

Historically, the OGTT has been used for the quantification of glycemia and screening for detection of diabetes [25]. However, understanding variables that relate to the regulation of glucose may lead to advances in the prediction of acute glycemic dysregulation (e.g., hyper or hypoglycemia). While our study demonstrated a relationship between FBG and HRV, when evaluating the AUC in relation to HRV metrics for the group, no significant correlations were observed (Table 3). These specific findings are similar to those of Rothberg et al. who also reported no relationship between resting HRV and post-prandial blood glucose values [23]. In contrast, Saito et al. reported that reductions in HRV were associated with lower insulin sensitivity, and a reduced insulin sensitivity index in non-overweight individuals [26].

Although we found no relationship between HRV and OGTT results in the full sample, we did see a significant correlation between HRV and glucose AUC in males (Fig 4), whereas female participants exhibited no such relationship. An independent samples t-test revealed no significant differences between male and female resting values of FBG, AUC and HRV metrics (Table 4). Though this study was not designed to evaluate the mechanisms related to our observations, it may be that the lack of a relationship between markers of HRV and AUC in females can be explained, in part, by hormonal fluctuations related to different stages in the menstrual cycle. Data collected support this possibility, as each female participant reported the first day of their last menstrual cycle and were on average 19.9 ± 11.5 days past the first day of their last menstrual cycle, making it plausible that various stages of the menstrual cycle were represented (Table 1). Brar et al. suggest that hormonal changes related to menstrual cycle alters ANS activity, with elevated SNS activity observed in later phases of the menstrual cycle compared to the predominant PSNS activity in earlier phases [27]. Future projects should account and control for menstrual cycle in order to determine the possible mechanism for sex-based differences.

## Limitations

Though this study was a novel attempt to evaluate glucose regulation, it was not without limitations. This study used a small sample size, which is an inherit limitation regardless of significant findings. Future studies should use a more robust sample size to better corollate physical and physiological contributors to HRV based OGTT relationship. Furthermore, this study was not designed to control for menstrual cycle. Future studies should enroll female participants in a fashion to control for stage of menstrual cycle. This study used single-day HRV to predict same-day metabolic function; though appropriate for comparing to same-day OGTT, this value may be stronger when compared to a weekly average, given the levels of variability we observed in these markers. Future studies should extend the number of measures glucose time points beyond the two-hour mark in order to capture the downward tail of the OGTT and capture full glucose clearance. The measurement of glucose concentration in this study was performed using finger-prick, future projects should use vein collection to reduce standard error. Lastly, plasma insulin may provide additional insight for this study.

## Conclusion

Heart rate variability was not predictive of oral glucose tolerance performance when participants were evaluated as a whole. However, a positive relationship was observed when the data were separated by sex, specifically in males. Though mechanisms were not identified, these findings provide insights into the relationship between autonomic activity and glucose uptake. In addition to already-established applications, HRV as an index of ANS activity may provide meaningful background and mechanistic information to that derived from metabolic assessments. Future research should control for menstrual cycle to determine if a relationship also exists in females.

## Author Contributions

**Conceptualization:** Abigail Nickel, Robert Buresh, Brian Kliszczewicz.

**Formal analysis:** Abigail Nickel, Robert Buresh, Cherilyn McLester, Brian Kliszczewicz.

**Funding acquisition:** Brian Kliszczewicz.

**Investigation:** Abigail Nickel, Robert Buresh, Andre Canino, Gabe Wilner, Keilah Vaughan, Pedro Chung, Brian Kliszczewicz.

**Methodology:** Robert Buresh, Brian Kliszczewicz.

**Project administration:** Abigail Nickel, Brian Kliszczewicz.

**Resources:** Robert Buresh, Brian Kliszczewicz.

**Supervision:** Abigail Nickel, Robert Buresh, Cherilyn McLester, Brian Kliszczewicz.

**Validation:** Robert Buresh, Cherilyn McLester, Brian Kliszczewicz.

**Visualization:** Abigail Nickel, Robert Buresh, Cherilyn McLester, Brian Kliszczewicz.

**Writing – original draft:** Abigail Nickel, Brian Kliszczewicz.

**Writing – review & editing:** Abigail Nickel, Robert Buresh, Cherilyn McLester, Andre Canino, Gabe Wilner, Keilah Vaughan, Pedro Chung, Brian Kliszczewicz.

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
