## [Decision Letter · Decision Letter 0]

30 Jan 2024

PONE-D-23-28922The relationship between heart rate variability and glucose clearance in healthy men and womenPLOS ONE

Dear Dr. Nickel,

Thank you for submitting your manuscript to PLOS ONE. After careful consideration, we feel that it has merit but does not fully meet PLOS ONE’s publication criteria as it currently stands. Therefore, we invite you to submit a revised version of the manuscript that addresses the points raised during the review process.

We look forward to receiving your revised manuscript.

Kind regards,

Hidetaka Hamasaki

Academic Editor

PLOS ONE

Journal Requirements:

Additional Editor Comments:

Thank you for submitting your valuable work to PLOS ONE.

The editor also has a concern for the small sample size.

I would appreciate it if you could address the issues raised by the reviewers.

Reviewers' comments:

Reviewer's Responses to Questions

**Comments to the Author**

1. Is the manuscript technically sound, and do the data support the conclusions?

Reviewer #1: Partly

Reviewer #2: Partly

2. Has the statistical analysis been performed appropriately and rigorously? 

Reviewer #1: Yes

Reviewer #2: Yes

3. Have the authors made all data underlying the findings in their manuscript fully available?

Reviewer #1: Yes

Reviewer #2: Yes

4. Is the manuscript presented in an intelligible fashion and written in standard English?

Reviewer #1: Yes

Reviewer #2: Yes

5. Review Comments to the Author

Reviewer #1: The authors have found significant correlations between heart rate variability (HRV) markers and fasting blood glucose levels in healthy individuals. However, they did not find any significant correlations between HRV markers and the area under the curve (AUC) of glucose. When analyzing by sex, only men showed significant correlations between AUC and RMSSD, SDNN, and LF. There were no sex differences for fasting blood glucose levels, AUC, RMSSD, SDNN, HF, and LF.

However, it is important to note that this is a small study with a limited sample size. a sample size of at least 30 is recommended for statistical tests because it is large enough to approximate the true distribution of the population being studied.

Was the HRV derived from ECG with your own algorithm, or it was just getting from the ECG monitor?

authors should explain why they chose the statistical measures.

The study you provided does not mention the use of a menstrual cycle survey in the correlations of measures, why did the authors performed that suevey, the same for the Urine Specific Gravity?

Reviewer #2: The study by Nickel et al. investigated the impact of oral glucose on the extent of HRV, considered an important biomarker for metabolic health. The participants were given a 75-g glucose load, and HRV was measured before its ingestion. Different HRV calculations/parameters were used to determine its association with the OGTT response (assessed as the AUC). The results showed that HRV parameters were correlated with capillary glucose concentration. Considering sex in the statistical model, correlations disappeared in women but not in men. Furthermore, HRV and AUC parameters were also correlated in the group of men. The research was properly conducted; methods are appropriate, and results might have certain implications regarding the HRV field. I have the following constructive comments that hopefully can further increase the quality of this manuscript.

Method section:

The study was carefully designed and conducted. However, the sample size was small (n=18; 10 men and 8 women). Could authors mention this issue (I missed this information in the Method section)? As the small sample size included in the study might impact data interpretation. Thank you.

Could the authors better explain what “apparently healthy individuals” is? Did authors determine that using blood samples, personal/medical interviews, questionnaires or is self-reported? Please provide more information as the paper is focused on “healthy individuals.”

Was glucose metabolism impairment established as an inclusion/exclusion criterion? If yes, how was it determined?

Regarding glucose concentration, authors should mention they are measuring “capillary blood glucose” instead of “blood glucose.”

Could authors provide the exact fasting time (e.g., mean and SD). Thank you.

Was a commercial glucose drink used, or "homemade" - please state the source and volume (this information was not provided).

Does the participant have an “acclimation period” prior to the HRV assessment, or did the HRV assessment start immediately after participants were placed in bed/stretcher?

I noticed authors used the low artifact correction filter in the Kubios software. Could the authors better explain why they used this filter? In a relatively recent study (PMID: 31979367), the authors proposed the medium filter for young adults. In this regard, did the authors check whether the results are independent of the Kubios filter used?

AUC calculation: I understood the rationale for computing the AUC using the Tai’s Mathematical Model; however, I wonder whether that AUC calculation provides glucose values as “arbitrary units” or “no units” (as certain AUC calculations [e.g., PMID: 14756916]). Could authors mention this aspect?

The thermic effect of a 75g bolus of glucose was unlikely to have been completed within the 3-hour measurement window - therefore it is possible that some differences in the response may have been missed by not capturing the downward tail of the HRV response curve. Should this not warrant a mention? Moreover, did capillary glucose concentration values return to baseline? If not, do the authors believe this issue could partially explain the results (e.g., absence of associations)? If capillary glucose did not return to baseline levels after 2-h, glucose-stimulation could be influencing the results. I would suggest mention this issue. Finally, could authors provide a figure showing the capillary glucose concentration and across time (i.e., the curve)? Thank you.

Does a protocol of repeated finger-prick collection add to within-subject variability? Capillary blood is notoriously difficult to standardize at each time point of collection, though it may relate to venous values.

Lines 133-135: No information regarding missing data is provided. Although Table 2 included 18 participants, I noticed that Figures (1 – 3) showed 15 participants instead of 18. Please mention this issue.

Why was the menstrual cycle recorded/registered if it was not used for analyses? Did the authors include the menstrual cycle in the analyses (e.g., as a confounder factor for correlations)? If not, I would suggest placing Table 1 as supplementary material or embed it in Table 2.

New analysis: I would suggest repeating the analyses, including heart rate (in beats per minute).

Discussion section:

Lines 173-175: This could be related to the low statistical power/sample size, rather than not an effect was observed. This issue deserves a mention.

Lines 179-181: This threshold is commonly used when blood glucose concentration (i.e., vein samples) is obtained. As I mentioned previously, I would recommend emphasizing that capillary instead of vein glucose concentration was assessed.

Limitation section:

In my opinion, the following should be included: The small sample size should be mentioned in this section. The menstrual cycle should be mentioned as a potential limitation, as women were at different menstrual cycle statuses which could impact the results in an unknown manner. Glucose concentration was assessed by a finger-prick instead of vein collection.

I hope you find these suggestions helpful for refining your manuscript.

Kind regards,

6. PLOS authors have the option to publish the peer review history of their article (what does this mean?). If published, this will include your full peer review and any attached files.

Reviewer #1: **Yes: **GILBERTO IVAN PERPINAN ISEDA

Reviewer #2: No

---

## [Author Response · Author response to Decision Letter 0]

27 Feb 2024

Reviewer #1: 

The authors have found significant correlations between heart rate variability (HRV) markers and

fasting blood glucose levels in healthy individuals. However, they did not find any significant

correlations between HRV markers and the area under the curve (AUC) of glucose. When

analyzing by sex, only men showed significant correlations between AUC and RMSSD, SDNN,

and LF. There were no sex differences for fasting blood glucose levels, AUC, RMSSD, SDNN,

HF, and LF.

However, it is important to note that this is a small study with a limited sample size. a sample

size of at least 30 is recommended for statistical tests because it is large enough to approximate

the true distribution of the population being studied.

The following was changed in the limitations:

“Though this study was a novel attempt to evaluate glucose regulation, it was not without

limitations. This study used a small sample size, which is an inherit limitation regardless of

significant findings. Future studies should use a more robust sample size to better corollate

physical and physiological contributors to HRV based OGTT relationship. Furthermore, this

study was not designed to control for menstrual cycle. Future studies should enroll female

participants in a fashion to control for stage of menstrual cycle. This study used single-day HRV

to predict same-day metabolic function; though appropriate for comparing to same-day OGTT,

this value may be stronger when compared to a weekly average, given the levels of variability

we observed in these markers. The measurement of glucose concentration in this study was

performed using finger-prick, future projects should use vein collection to reduce standard error.

Lastly, plasma insulin may provide additional insight for this study.”

1) Was the HRV derived from ECG with your own algorithm, or it was just getting from the

ECG monitor?

HRV was derived using Kubios software (version 3.0.5). The files collected from the Finapres

NOVA were transferred to Kubios for analysis. The following was edited in the methods to

clarify this more.

“Measurements for the current study were collected using the Finapres NOVA, and ECG files

were transferred to Kubios software (version 3.0.5).”

2) authors should explain why they chose the statistical measures.

“We used parametric stats appropriate for the nature of our data.   Because parametric stats

assume that data are distributed normally, performed a test for normality, to ensure that our data

were indeed distributed normally.  We performed two-tailed correlation analysis to be

conservative in our conclusions. Likewise, we did a two-tailed independent samples t-test to

conservatively compare results between sexes.”

3) The study you provided does not mention the use of a menstrual cycle survey in the

correlations of measures, why did the authors performed that survey, the same for the Urine

Specific Gravity?

Menstrual cycle information was not entered into the correlation due to the low sample size and

was used as observational data. Previous work in our lab suggested a relationship between HRV

and menstrual cycle, so we collected the phase data as a participant characteristic in the event we

observed a sex-based difference. USG was collected for the BIA hydration

Reviewer #2: 

The study by Nickel et al. investigated the impact of oral glucose on the extent of HRV,

considered an important biomarker for metabolic health. The participants were given a 75-g

glucose load, and HRV was measured before its ingestion. Different HRV

calculations/parameters were used to determine its association with the OGTT response

(assessed as the AUC). The results showed that HRV parameters were correlated with capillary

glucose concentration. Considering sex in the statistical model, correlations disappeared in

women but not in men. Furthermore, HRV and AUC parameters were also correlated in the

group of men. The research was properly conducted; methods are appropriate, and results might

have certain implications regarding the HRV field. I have the following constructive comments

that hopefully can further increase the quality of this manuscript.

Method section:

The study was carefully designed and conducted. However, the sample size was small (n=18; 10

men and 8 women). Could authors mention this issue (I missed this information in the Method

section)? As the small sample size included in the study might impact data interpretation. Thank

you.

This information we added to the limitation section.

Could the authors better explain what “apparently healthy individuals” is? Did authors determine

that using blood samples, personal/medical interviews, questionnaires or is self-reported? Please

provide more information as the paper is focused on “healthy individuals.”

“Inclusion and exclusion criteria were determined through self-report through the health history

questionnaire”

Was glucose metabolism impairment established as an inclusion/exclusion criterion? If yes, how

was it determined?

Known metabolic conditions were a part of the exclusion criteria and determined through the

self-report HHQ.

Regarding glucose concentration, authors should mention they are measuring “capillary blood

glucose” instead of “blood glucose.”

Fixed

Could authors provide the exact fasting time (e.g., mean and SD). Thank you.

We only confirmed that they had not eaten prior to 12 hours before arriving to the lab, so we do

not have exact times.

Was a commercial glucose drink used, or "homemade" - please state the source and volume (this

information was not provided).

OGTT beverage containing 75 grams of glucose (Trutol TM )

Does the participant have an “acclimation period” prior to the HRV assessment, or did the HRV

assessment start immediately after participants were placed in bed/stretcher?

The following was added:

“on an examination bed” was added to the Experimental Design section.

“The first five minutes of the 10-minute recording were discarded for acclimatation. The last

five minutes of resting time points were analyzed.”

I noticed authors used the low artifact correction filter in the Kubios software. Could the authors

better explain why they used this filter? In a relatively recent study (PMID: 31979367), the

authors proposed the medium filter for young adults. In this regard, did the authors check

whether the results are independent of the Kubios filter used?

We visually inspected each recording for artifact when applying the filters and used the lowest

filter setting possible to avoid augmenting the recording beyond the natural variations that may

occur. The citations we used warned against too strong of filters as it may “over filter” the

recording and lose natural variation. The following was added for clarification:

The Kubios “low artifact correction” filter with a � 0.35 sec sensitivity to R-R abnormalities

compared to the local average was used [12,13].

AUC calculation: I understood the rationale for computing the AUC using the Tai’s

Mathematical Model; however, I wonder whether that AUC calculation provides glucose values

as “arbitrary units” or “no units” (as certain AUC calculations [e.g., PMID: 14756916]). Could

authors mention this aspect?

The reviewer brings up a good point, the unit used in this model was “arbitrary units”, this has

been added into the manuscript and is expressed as AU.

The thermic effect of a 75g bolus of glucose was unlikely to have been completed within the 3-

hour measurement window - therefore it is possible that some differences in the response may

have been missed by not capturing the downward tail of the HRV response curve. Should this not

warrant a mention?

The reviewer makes a good point. The duration of the OGTT used was the standard two hours.

However, we added the following to the limitation section:

Future studies should extend the number of measures glucose time points beyond the two-hour

mark in order to capture the downward tail of the OGTT and capture full glucose clearance.

Moreover, did capillary glucose concentration values return to baseline? If not, do the authors

believe this issue could partially explain the results (e.g., absence of associations)? If capillary

glucose did not return to baseline levels after 2-h, glucose-stimulation could be influencing the

results. I would suggest mention this issue.

Capillary glucose concentration values did not return to baseline (Figure 4). We believe this was

addressed with the change we made with the previous comment.

Finally, could authors provide a figure showing the capillary glucose concentration and across

time (i.e., the curve)? Thank you.

Figure 4 was added for capillary glucose over time

Does a protocol of repeated finger-prick collection add to within-subject variability? Capillary

blood is notoriously difficult to standardize at each time point of collection, though it may relate

to venous values.

This is a good comment and we added the limitation of finger sticks with future work utilizing

venipuncture.

Lines 133-135: No information regarding missing data is provided. Although Table 2 included

18 participants, I noticed that Figures (1 – 3) showed 15 participants instead of 18. Please

mention this issue.

Three participants were dropped due to ECG error, they were removed from analysis and we

accidentally left the total group in the table. Table 2 has been updated and the following was

added:

“Eighteen apparently healthy volunteers completed the study: ten males and eight females. Two

males and one female were removed from data analysis due to artifact or ECG recording error,

15 total participants were analyzed, eight males and seven females.”

Why was the menstrual cycle recorded/registered if it was not used for analyses? Did the authors

include the menstrual cycle in the analyses (e.g., as a confounder factor for correlations)? If not, I

would suggest placing Table 1 as supplementary material or embed it in Table 2.

Menstrual cycle information was not entered into the correlation due to the low sample size and

was used as observational data. Previous work in our lab suggested a relationship between HRV

and menstrual cycle, so we collected the phase data as a participant characteristic in the event we

observed a sex-based difference.

New analysis: I would suggest repeating the analyses, including heart rate (in beats per minute).

RHR correlation data was added to table 3 and results.

Discussion section:

Lines 173-175: This could be related to the low statistical power/sample size, rather than not an

effect was observed. This issue deserves a mention.

Sample size was added as a limitation to the limitation section.

Lines 179-181: This threshold is commonly used when blood glucose concentration (i.e., vein

samples) is obtained. As I mentioned previously, I would recommend emphasizing that capillary

instead of vein glucose concentration was assessed.

The following was edited:

“The participants in this study were considered healthy, normal weight individuals with an

average capillary FBG of 89.9 ± 8.2 mg/dL.”

Limitation section:

In my opinion, the following should be included: The small sample size should be mentioned in

this section. The menstrual cycle should be mentioned as a potential limitation, as women were

at different menstrual cycle statuses which could impact the results in an unknown manner.

Glucose concentration was assessed by a finger-prick instead of vein collection.

The following was changed in the limitations:

“Though this study was a novel attempt to evaluate glucose regulation, it was not without

limitations. This study used a small sample size, which is an inherit limitation regardless of

significant findings. Future studies should use a more robust sample size to better corollate

physical and physiological contributors to HRV based OGTT relationship. Furthermore, this

study was not designed to control for menstrual cycle. Future studies should enroll female

participants in a fashion to control for stage of menstrual cycle. This study used single-day HRV

to predict same-day metabolic function; though appropriate for comparing to same-day OGTT,

this value may be stronger when compared to a weekly average, given the levels of variability

we observed in these markers. The measurement of glucose concentration in this study was

performed using finger-prick, future projects should use vein collection to reduce standard error.

Lastly, plasma insulin may provide additional insight for this study.”

---

## [Decision Letter · Decision Letter 1]

2 Apr 2024

PONE-D-23-28922R1The relationship between heart rate variability and glucose clearance in healthy men and womenPLOS ONE

Dear Dr. Nickel,

Thank you for submitting your manuscript to PLOS ONE. After careful consideration, we feel that it has merit but does not fully meet PLOS ONE’s publication criteria as it currently stands. Therefore, we invite you to submit a revised version of the manuscript that addresses the points raised during the review process.

We look forward to receiving your revised manuscript.

Kind regards,

Hidetaka Hamasaki

Academic Editor

PLOS ONE

Journal Requirements:

Reviewers' comments:

Reviewer's Responses to Questions

**Comments to the Author**

1. If the authors have adequately addressed your comments raised in a previous round of review and you feel that this manuscript is now acceptable for publication, you may indicate that here to bypass the “Comments to the Author” section, enter your conflict of interest statement in the “Confidential to Editor” section, and submit your "Accept" recommendation.

Reviewer #2: All comments have been addressed

2. Is the manuscript technically sound, and do the data support the conclusions?

Reviewer #2: Yes

3. Has the statistical analysis been performed appropriately and rigorously? 

Reviewer #2: Yes

4. Have the authors made all data underlying the findings in their manuscript fully available?

Reviewer #2: Yes

5. Is the manuscript presented in an intelligible fashion and written in standard English?

Reviewer #2: Yes

6. Review Comments to the Author

Reviewer #2: I would like to thank the authors for taking the time to make revisions based on my comments and suggestions. Please find below my minor comments and suggestions:

1. I would suggest adding the lack of an exact fasting time to the limitations section, as the authors 'recommended' fasting for ≥ 12 hours, but cannot confirm it.

2. References 12 and 13 are cited to justify Kubios artifacts 'over filter' correction. I would suggest including the following citation (PMID: 31979367) as it addresses this issue.

3. How resting heart rate (RHR) was derived from the ECG signal should be added to the methods section. Additionally, please check the units of RHR in Table 3 (should be bpm instead of bmp).

4. I noticed that information concerning detrending, lambda, and interpolation rate was not provided in the Kubios software description.

5. In the Methods section, when referring to the Fast Fourier Transformation technique, I would suggest changing 'technique' to 'algorithm' for clarity.

7. PLOS authors have the option to publish the peer review history of their article (what does this mean?). If published, this will include your full peer review and any attached files.

Reviewer #2: No

---

## [Author Response · Author response to Decision Letter 1]

22 Apr 2024

We have addressed the most recent revisions in the newest 'Reviewer comments' attachment: 

We re-examined the data files and found the participant fasting times were Total - 13.1 +/- 1.8 hours, Males - 12.6 +/- 2.1 hours, and Females

- 13.7 +/- 1.2 hours.

The following citation was added:

Alcantara JMA, Plaza-Florido A, Amaro-Gahete FJ, Acosta FM, Migueles JH, Molina-Garcia P,

et al. Impact of Using Different Levels of Threshold-Based Artefact Correction on the

Quantification of Heart Rate Variability in Three Independent Human Cohorts. J Clin Med. 2020

Jan 23;9(2):325.

The following was changed:

Methods: HRV Analysis section

The last five minutes of resting time points were analyzed for HRV and RHR.

Bmp was changed to bpm.

Detrending was set to smoothing priors with a smoothing parameter of 500, lambda of 0.035 Hz,

and Interpolation rate of 4 Hz

---

## [Editor Report · Decision Letter 2]

24 Apr 2024

The relationship between heart rate variability and glucose clearance in healthy men and women

PONE-D-23-28922R2

Dear Dr. Kliszczewicz,

We’re pleased to inform you that your manuscript has been judged scientifically suitable for publication and will be formally accepted for publication once it meets all outstanding technical requirements.

Kind regards,

Hidetaka Hamasaki

Academic Editor

PLOS ONE
---

## [Editor Report · Acceptance letter]

14 May 2024

PONE-D-23-28922R2 

PLOS ONE

Dear Dr. Kliszczewicz, 

I'm pleased to inform you that your manuscript has been deemed suitable for publication in PLOS ONE. Congratulations! Your manuscript is now being handed over to our production team.

Kind regards, 

on behalf of

Dr. Hidetaka Hamasaki 

Academic Editor

PLOS ONE